# Cytotoxicity and Biomineralization Potential of Flavonoids Incorporated into PNVCL Hydrogels

**DOI:** 10.3390/jfb14030139

**Published:** 2023-03-02

**Authors:** Gabriela Pacheco de Almeida Braga, Karina Sampaio Caiaffa, Rafaela Laruzo Rabelo, Vanessa Rodrigues dos Santos, Amanda Caselato Andolfatto Souza, Lucas da Silva Ribeiro, Emerson Rodrigues de Camargo, Anuradha Prakki, Cristiane Duque

**Affiliations:** 1Department of Preventive and Restorative Dentistry, Araçatuba Dental School, São Paulo State University (UNESP), Araçatuba 16015-050, SP, Brazil; 2Department of Chemistry, Federal University of São Carlos (UFSCar), São Carlos 13565-905, SP, Brazil; 3Dental Research Institute, Faculty of Dentistry, University of Toronto, Toronto, ON M5G 1G6, Canada

**Keywords:** flavonoids, hydrogel, odontoblasts, cytotoxicity, alkaline phosphatase, dentin mineralization

## Abstract

This study aimed to evaluate the effects of flavonoids incorporated into poly(N-vinylcaprolactam) (PNVCL) hydrogel on cell viability and mineralization markers of odontoblast-like cells. MDPC-23 cells were exposed to ampelopsin (AMP), isoquercitrin (ISO), rutin (RUT) and control calcium hydroxide (CH) for evaluation of cell viability, total protein (TP) production, alkaline phosphatase (ALP) activity and mineralized nodule deposition by colorimetric assays. Based on an initial screening, AMP and CH were loaded into PNVCL hydrogels and had their cytotoxicity and effect on mineralization markers determined. Cell viability was above 70% when MDPC-23 cells were treated with AMP, ISO and RUT. AMP showed the highest ALP activity and mineralized nodule deposition. Extracts of PNVCL+AMP and PNVCL+CH in culture medium (at the dilutions of 1/16 and 1/32) did not affect cell viability and stimulated ALP activity and mineralized nodules’ deposition, which were statistically higher than the control in osteogenic medium. In conclusion, AMP and AMP-loaded PNVCL hydrogels were cytocompatible and able to induce bio-mineralization markers in odontoblast-cells.

## 1. Introduction

The physiological process of apexogenesis, which consists of normal root development and closure of the apex mainly through the deposition of dentin by odontoblasts, can be interrupted in young permanent teeth by the occurrence of trauma or pulp necrosis. These teeth generally are treated with a conventional apexification procedure that induces the formation of a mineralized barrier in the apical region, which is based on frequent exchanges of calcium hydroxide (CH) [1]. This procedure induces root weakening and increases the susceptibility to root fracture [1,2], in addition to providing little or no benefit to the continuity of apicigenesis [3]. Another material used for apexification is mineral trioxide aggregate (MTA). MTA has demonstrated in vivo osteoinductive properties by inducing dynamic biomineralization processes in surgical bone defects [4]. The apical plug created in the most apical portion of the immature root with MTA can seal the communication between the root canal and the extra-radicular region, showing a greater advantage over frequent exchanges of CH [5]. However, this technique depends on the crown/root ratio, and it does not promote complete root development, leaving a risk of further root fracturing [5]. Currently, the American Association of Endodontics (AAE) has also recommended the regenerative endodontic procedure (REP) for endodontic treatment of young permanent teeth as an alternative to apexification [6]. REP consists of two sessions: (1) careful disinfection the root canal with sodium hypochlorite at low concentrations, and insertion of an antimicrobial medication, such as antibiotic pastes or calcium hydroxide pastes, for 2–3 weeks; and (2) reopening, blood clot induction, MTA/CH plugging and restoration [6] Although satisfactory clinical and radiographic results were observed, with most of cases exhibiting complete root development [6], histological evaluations of REP have shown the apposition of a connective tissue associated with cementum or bone, and no evidence of dentin deposition suggesting a tissue repair, so not true regeneration. [7]

Considering the limitations of REP, new cell-based and cell-free approaches for regeneration of the pulp–dentin complex have arisen. In cell-based methods, allogenic stem cells (derived from the host) are seeded in scaffolds and inserted into root canals. A cell-free approach uses biomolecules in an attempt to stimulate biomineralization by the remaining cells or induce differentiation of endogenous stem cells [8]. This technique is simpler, lower cost and requires less training by clinicians than cell-based methods; however, there is not enough knowledge about which biomolecules could be applied for endodontic regeneration [8]. Several biomolecules have been screened for their potential in inducing differentiation, proliferation and mineralization biomarkers in odontogenic/osteogenic cells [9].

Flavonoids are polyphenolic compounds widely found in various fruits, vegetables, barks, stems, tea plants and derivatives. They are largely used in various sectors ranging from nutritional and pharmaceutical to medicinal and cosmetics [10]. Flavonols and derivates, such as dihydroflavonols and glycosylated flavonols, are groups of flavonoids known for their strong antimicrobial and antioxidant activity, and thereby their multiple benefits to human health [11,12]. Ampelopsin (AMP), also called dihydromyricetin, is a flavonoid extracted mainly from the leaves of *Ampelopsis grossendata*. Some studies have shown its biological and pharmacological properties, such as anti-inflammatory, antioxidant, anti-tumor, hepatoprotective, cardioprotective, neuroprotective, dermoprotective, insulin and cholesterol regulating and antimicrobial activities [13,14,15]. AMP also had effects on bone mesenchymal stem cells’ osteogenesis: increasing ALP activity, osteoblast-specific gene expression and mineral deposition [16]. Isoquercitrin (quercetin-3-O-glucoside) is a flavonoid glucoside found in plants, fruits and vegetables that has antioxidant, anti-inflammatory [17], anti-cancer [18] and antiviral [19] properties. Furthermore, it has been demonstrated to enhance the mineralization capacity of osteoblastic cells and to promote alkaline phosphatase activity [20,21]. Rutin is another glycosylated flavonol found in teas; fruits such as apples and tomatoes; and legumes such as onions, among others [22]. Rutin has also demonstrated antioxidant, anti-inflammatory and anticancer activities in some recent studies [23,24]. Studies observed the effects of rutin on cell proliferation and reported that it enhanced osteogenic differentiation and mineralization and reduced oxidative stress [25,26].

To improve their solubility, bioavailability and biological properties in a controlled-release manner in an attempt to eliminate residual bacteria, inflammation and the reparative process, flavonoids have been incorporated into drug-delivery vehicles, such as hydrogels [27,28]. The poly(n-vinylcaprolactam) (PNVCL) hydrogel is a thermoreversible hydrogel capable of transitioning from the liquid state to a gel when subjected to a temperature near the physiological human body temperature. It becomes a gel at approximately 34 °C and returns to its liquid form at lower temperatures [29,30]. It has shown great potential in the field of biomedicine as a controlled drug delivery system by being easily injectable and biocompatible, and by not producing toxic compounds and increasing the availability of drugs [29,30,31,32,33,34].

In endodontics, the search for a medication that can remain inside the root canal for long periods of time, eliminate the residual microorganisms and stimulate the remaining odontoblastic cells or progenitor cells of the apical papilla to continue root development is still a challenge [3]. In a recent study published by our research group, AMP-loaded PNVLC hydrogels demonstrated low cytotoxicity and an antimicrobial effect against bacteria related to endodontic infections [34]. Considering these previous findings, the present study aimed to evaluate the effects of flavonoids (ampelopsin, isoquercitrin and rutin) and an AMP-loaded PNVCL hydrogel on the viability and biomineralization potential of odontoblast-like cells, in the search for a potential injectable delivery system with multifunctional activities for application as an endodontic medication. The null hypothesis of this study was that there are no differences (1) among flavonoids and (2) among PNVCL hydrogels containing a flavonoid or not, or CH (as control), in terms of their cytotoxicity and mineralization potential.

## 2. Materials and Methods

### 2.1. Preparation of Compounds and Controls

Ampelopsin (AMP, #42866), isoquercitrin (ISO, #17793) and rutin (RUT, #R5143 were dissolved in dimethyl sulfoxide (DMSO, Sigma-Aldrich, St. Louis, MO, USA) at 30 mg/mL and stored at −70 °C. Calcium hydroxide was dissolved in water at 1 mg/mL and stored at 4 °C. All compounds were obtained from Sigma-Aldrich (St. Louis, MO, USA). Cell culture medium was used as a negative control in this study. All experiments were performed in triplicate in three independent experiments (*n* = 9).

### 2.2. Synthesis and Characterization of PNVCL Hydrogels

Thermosensitive PNVCL hydrogels were synthesized and characterized beforehand by ^1^H NMR, FITR and ultrastructural images [29,30,31,32,33,34]. Briefly, N-vinylcaprolactam monomer (NVCL) was dissolved in dimethyl sulfoxide (DMSO) and heated to 70 °C under a N_2_ atmosphere. Afterward, azobisisobutyronitrile dissolved in DMSO was added to the system and kept under agitation for 4 h. Finally, PNVCL was purified by dialysis for 4 days, dried at 50 °C and stored at 4 °C. For subsequent assays, PNVCL hydrogel was loaded with AMP at 2.5 mg/mL (7.8 mM, PNVCL+AMP) and CH at 1 mg/mL (13.5 mM, PNVCL+CH) separately and incubated for 24 h at 5 °C for total solubilization and preparation of a polymeric gel as previously described [34]. Rheological analysis and determination of compounds (AMP and CH) released from PNVCL hydrogels were recently published [34].

### 2.3. MDP-23 Cell Culture

The immortalized odontoblast-like cells (MDPC-23—mouse dental papilla cell line) were obtained from Prof. Dr. Carlos Alberto de Souza Costa (FOAr-UNESP, Araraquara, Brazil). They were cultured in Dulbecco’s Modified Eagle’s Medium (DMEM; high low glucose, L-glutamine, and sodium pyruvate; Gibco, Grand Island, NY, USA) supplemented with 10% fetal bovine serum (FBS; Gibco) and containing 100 IU/mL penicillin, 100 µg/mL streptomycin, 100 µg/mL gentamicin and 0.25 µg/mL amphotericin (Gibco) in a humidified incubator with 5% CO_2_ and 95% air at 37 °C (Isotemp Fisher Scientific, Pittsburgh, PA, USA). Cells were sub-cultured every 2 days until 80% confluence was reached [35].

### 2.4. Study Design

The experimental design of the study is presented in Figure 1. For the first part of this study (Figure 1A), the samples were solutions of the following compounds: ampelopsin (AMP), isoquercitrin (ISO), rutin (RUT) and calcium hydroxide (CH, gold standard as medication in endodontics) at concentrations of 100, 50 and 25 µM. The control group was DMEM medium (no treatment). MDPC-23 were seeded (5 × 10^3^ cells/well) onto sterile 96-well plates (T0), which were maintained at 37 °C for 48 h. Then, cells were treated with the compounds (T1) for 24 h (T2) and 48 h (T3) for cell viability determination [34,35]. For alkaline phosphatase (ALP) and mineralized nodules (MN) deposition assays, cells were seeded in 48 well plates (3 × 10^2^ cells/well) and treated for 48 h with the compounds (T3). After that, DMEM was replaced by osteogenic DMEM (DMEM with FBS, antibiotics and supplemented with 50 μg/mL ascorbic acid, 10 nmol/L sodium β-glycerophosphate and 1.8 nmol/L KH_2_PO_4_, which was refreshed every 48 h until completion of the experimental period for ALP (8 days—T4) and MN deposition (14 days—T5) assays (Figure 1A). For the second part of this study (Figure 1B), the samples were serial dilutions of the PNVCL hydrogel extracts (PNVCL, PNVCL+AMP and PNVC+CH). One milliliter of each PNVCL hydrogel at the liquid state was inserted in 24-well plates and incubated overnight at 37 °C to acquire the gel state [34]. Then, 1 mL of DMEM was added over the hydrogels (T0) and incubated for 48 h (T1) and 7 days (T3) to obtain the hydrogel extracts. Cells were seeded (T2) and evaluated for cytotoxicity at 48 h (T4), 8 days (T5) and 14 days (T6); for ALP at 8 days (T5); and for MN deposition at 14 days (T6) (Figure 1B). All assays were performed in triplicate in three independent experiments [36].

### 2.5. Cell Viability Assay

The cell viability was evaluated using resazurin colorimetric assays at 48 h, 8 days and 14 days after treatments, according to previous studies [34,35,36,37]. Briefly, cells were seeded into 96-well plates (5 × 10^3^ cells/well) and incubated for 48 h. For flavonoid screening, culture medium was aspirated and cells were treated with AMP, ISO and RUT at 100, 50 and 25 µM for 48 h under standard cell culture conditions. After incubation, treatments were aspirated, and cells were maintained in osteogenic DMEM (changed every 48 h) until completing 8 and 14 days. For hydrogels evaluation, 48 h and 7 days extracts (diluted from 1/2 to 1/64) of PNVCL, PNVCL+AMP and PNVCL+CH were applied to the cells, and they were incubated for 48 h. For the subsequent assays, cells were treated with the PNVCL extracts diluted at 1/16 and 1/32 for 48 h, followed by osteogenic DMEM changes until completing 8 and 14 days. After all these periods, cells were washed with sodium phosphate buffer (PBS, 10 mM, pH 6.8) and stained with resazurin (#7017, Sigma Aldrich) for 4 h. Absorbance values were read at each time point at 570 and 600 nm in a spectrophotometer (Biotek, Winooski, VT, USA). These values were converted into percentages of cell viability considering DMEM medium as 100% of growth [34,35,36,37].

#### 2.5.1. Determination of Total Protein and Alkaline Phosphatase Activity

MDPC-23 at 3 × 10^2^ cells/well in 48-wells plates were treated with flavonoids (from 25 to 100 μM) or the PNVCL extracts (diluted at 1/16 and 1/32) for 48 h with subsequent osteogenic DMEM changing until completing 8 days. Total protein (TP) quantification and alkaline phosphatase activity (ALP) assays were conducted according to previous studies [35,36,37]. Briefly, cells were washed after aspiration of cell culture, and 0.1% sodium lauryl sulfate (Sigma-Aldrich) was added to each well for 10 min to lyse cells. For TP determination, Lowry reagent (Sigma-Aldrich) was added to 100 μL of the lysed cells. They were then incubated for 20 min, which was followed by adding Folin–Ciocalteu’s phenol reagent (Sigma-Aldrich) for 30 min. Afterward, samples were read in a spectrophotometer at 655 nm. A standard curve of bovine serum albumin (BSA, Sigma-Aldrich) was used to determine the total protein of each sample in μg/mL. The alkaline phosphatase (ALP) assay was performed following the instructions of the ALP kit manufacturer (Labtest Diagnóstico S.A., Lagoa Santa, MG, Brazil). The cell lysate was mixed with thymolphthalein monophosphate and incubated for 15 min at 37 °C. After that, 2 mL of color reagent was added to each sample, which was then homogenized and had its absorbance measured at 590 nm. ALP activity was calculated based on a standard curve with known enzyme concentrations. The final ALP data (percentages in relation to the osteogenic DMEM control) were divided by the values of total protein (percentages in relation to the osteogenic DMEM control) to normalize the ALP results [35,36,37].

#### 2.5.2. Mineralized Nodule Deposition

Mineralized nodule deposition was determined by the alizarin red staining method according to previous studies [35,36,37]. Briefly, MDPC-23 at 2 × 10^3^ cells/well were treated with the flavonoids or the PNVCL extracts (diluted at 1/16 and 1/32) for 48 h, and osteogenic DMEM was changed until 14 days were complete. After that, cells were washed with PBS and fixed in 70% cold ethanol for 2 h. After that, 40 mmol/L Alizarin Red S (Sigma-Aldrich) was applied to the cells for 20 min under gentle agitation. Then, cells were washed twice with distilled water and allowed to dry. Stained cultures were photographed using an inverted microscope (Olympus BX51; Olympus, Miami, FL, USA). After capturing the images, 10% of cetylpyridinium chloride solution was added to the wells and left under agitation for 15 min. One-hundred microliters of each well was transferred to a microplate reader, and the absorbance values were obtained at 562 nm. The results were converted into percentages, considering the osteogenic DMEM control as 100%. The final mineralization nodule deposition data (percentages in relation to the osteogenic DMEM control) were divided by the values of cell viability obtained by the resazurin method (percentages in relation to the osteogenic DMEM control to normalize the results [35,36,37].

### 2.6. Statistical Analysis

Data from cytotoxicity assays, ALP activity and mineralized nodule deposition are expressed as mean ± standard deviation. After validating the homogeneity and homoscedasticity using Shapiro–Wilk and Leve tests, data were submitted to parametrical statistical methods: ANOVA (one or two-way) and post-hoc Tukey (for comparison among the groups) or Dunnett (for comparison between groups and control) tests. SPSS 19.0 software (SPSS Inc., Chicago, IL, USA) was used to run the statistical analysis, considering *p* < 0.05.

## 3. Results

### 3.1. Flavonoid Treatments

#### 3.1.1. Cell Viability

Figure 2 shows the percentages of cell viability obtained by resazurin colorimetric assays after exposure of MDPC-23 odontoblastic-like cells to AMP, ISO and RUT at three different time points, 48 h, 8 days and 14 days, normalized by the positive control (DMEM). At all-time points and concentrations tested, cell viability was above 70% when cells were treated with AMP, ISO or RUT. At 48 h, cell viability was dose-dependent only for RUT groups (Figure 2A). On day 8, there was no difference among the concentrations, considering each flavonoid separately (Figure 2B). After 14 days, ISO at 100 and 50 μM stimulated cell growth of 109 and 104%, respectively, and the growth statistically differed from that of AMP at 25 μM and RUT at 50 and 25 μM (Figure 2C).

#### 3.1.2. Alkaline Phosphatase Activity

Figure 3 shows means and standard deviations of ALP activity when MDPC-23 were exposed to flavonoids for 48 h and evaluated after 8 days. Among the flavonoids tested, AMP at 100 μM stimulated the greatest ALP activity: 1.85-fold more than the control. The activity level was different from what was stimulated by AMP at 50 μM and 25 μM (1.37- and 1.25-fold, respectively). ISO at 50 μM and RUT at 100 μM increased ALP activity 1.36- and 1.34-fold in MDCP-23 cells, compared to the control. CH increased ALP activity by between 2.23 and 1.55 times superior to the control, showing the highest results in the study.

#### 3.1.3. Mineralized Nodule Deposition

Figure 4 presents the means and standard deviations of mineralized nodule (NM) deposition after 48 h of MDPC-23 exposure to flavonoids and evaluation at 14 days. Statistical difference from the control was observed for all concentrations of AMP, RUT and CH, which increased 1.54- to 2.13-fold, 1.58- to 1.78-fold, and 1.57- to 1.88-fold the NM deposition after treatments. ISO at 25 μM differed from the control, increasing the NM deposition 1.67-fold. Appendix A shows representative microscopic images of alizarin staining for each group in this study. High levels of NM deposition can be seen for all groups at the lowest concentrations.

### 3.2. Hydrogels Extract Treatments

In general, among the flavonoids tested, AMP showed superior ALP activity and mineralized nodule deposition, differing from the control at their highest concentrations. Therefore, AMP was chosen to be incorporated in PNVCL for cytotoxicity determination and determination of its effects on mineralization markers in comparison to PNVCL+CH.

#### 3.2.1. Cell Viability Evaluation

Figure 5, Figure 6 and Figure 7 show the percentages of cell viability after MDPC-23 exposure for 48 h and 7 days to hydrogel extracts, and evaluation at 48 h, 8 days and 14 days after treatments. The 48 h extracts of PNVCL+AMP and PNVCL+CH hydrogels did not affect cell viability. PNVCL was cytocompatible (more than 80% cell viability) at a 1/8 dilution (Figure 5A). Figure 5B shows the results after treatment with 7-day PNVCL extracts. PNVCL, PNVCL+AMP and PNVCL+CH were not cytotoxic at 1/8 dilutions. After 8 days, cell viability remained above 90% for all 48 h extracts (Figure 6A) and 7-day extracts at concentrations of 1/16 and 1/32 (Figure 6B). Figure 7A,B represent cell viability after 14 days. All 48 h and 7-day PNVCL extracts (PNVCL, PNVCL+AMP and PNVCL+CH at 1/16 and 1/32 dilutions) were not cytotoxic, since percentage values are over 70–80%.

#### 3.2.2. Alkaline Phosphatase Activity after Treatment with Hydrogel Extracts

The alkaline phosphatase activity of MDPC-23 exposed to 48 h extracts and 7-day extracts of the hydrogels can be seen in Figure 8. For 48 h hydrogel extracts, there was an increase on ALP activity for both groups—PNVCL+AMP (1.30- to 1.34-fold) and PNVCL+CH (1.27- to1.43-fold), but there was difference between the dilutions (1/16 and 1/32) (Figure 8A). For 7-day hydrogel extracts, both dilutions of the PNVCL+AMP group promoted an increase in ALP activity (1.35–1.39-fold), there being no difference between them. Both levels were statistically different from the control level. The highest ALP activity was observed for PNVCL+CH at the dilution of 1/16 (2.1-fold), which was different from the 1/32 dilution (1.37-fold) and all other groups of the study (Figure 8B). PNVCL alone did not differ from the control (osteogenic DMEM).

#### 3.2.3. Mineralized Nodule Deposition of Extracts

Figure 9 shows the effect of hydrogel extracts on mineralized nodule (MN) deposition by MDPC-23 cells. For 48 h hydrogel extracts, the 1/32 dilution of PNVCL+AMP (2.6-fold) induced higher MN deposition than PNVCL+CH at both dilutions (1.97- to 2.23-fold) (Figure 9A). For 7-day hydrogel extracts, PNVCL+AMP at 1/32 (2.9-fold) presented the highest values of NM deposition, which were not significantly different from those of the 1/16 dilution (2.24-fold). PNVCL+CH at 1/32 (1.78-fold) and at 1/16 (1.47-fold) had similar results and differed statistically from the control (Figure 9B). PNVCL hydrogel did not influence NM deposition by cells. Appendix A show representative microscopic images of alizarin staining for each group of hydrogel extracts. The greatest nodule deposition occurred in the PNVCL+AMP 1/32 group and both dilutions of PNVCL+CH (1/16 and 1/32).

## 4. Discussion

The null hypothesis of this study was rejected, since the results obtained by AMP and AMP-loaded PNVCL hydrogel were different from the controls and other experimental groups.

Considering the wide range of therapeutic properties of the flavonoids and the search for new biomolecules for regenerative endodontic applications, this study evaluated the cytotoxicity and effects of three flavonol derivatives—ampelopsin (AMP), or dihydromyricetin; and the glycosylated flavonols isoquercitrin (ISO) and rutin (RUT)—on mineralization markers of odontoblast-like cells. All flavonols were cytocompatible when applied to MDPC-23 at the concentrations tested (between 100 and 25 μM), keeping cell viability above 70%, independent on the timepoint evaluated. AMP has demonstrated cytoprotective effects on porcine intestinal columnar epithelial cells, IPEC-J2, at 20 and 40 μM, when these cells were exposed to the mycotoxin deoxynivalenol [38], and increased HaCaT cells’ viability and suppressed UVA-induced production of inflammatory cytokines [39]. ISO and RUT also acted as neuroprotective agents on pheochromocytoma PC12 cells exposed to 6-hydroxydopamine at concentrations between 10 and 100 μM [40,41]. Rutin has also been demonstrated to be a cytoprotective flavonoid capable of attenuating the inflammatory response in macrophage cells [42], keratocytes and fibroblasts [43].

Alkaline phosphatase (ALP) and mineralized nodule deposition are considered useful markers of biomineralization in osteoblast and odontoblast cells [44]. Alkaline phosphatase (ALP) activity is the initial phase of the dentin matrix biomineralization, and the formation of mineralization nodules is the final product of cell differentiation [45]. In our study, all flavonoids induced ALP activity and mineralized nodule formation, at specific concentrations. Among the flavonoids tested, AMP had the greatest effect on mineralization markers, which differed from the control levels after using AMP’s highest concentrations. AMP also exhibited no cytotoxic effect on human bone narrow mesenchymal stem cells (BMSCs) from 0.1 to 50 μM and increased ALP activity, osteoblast-specific gene expression and mineral deposition, revealing enhanced osteogenic differentiation [16]. In another study, RUT induced cell proliferation from 0.1 to 100 μM concentrations and protected periodontal ligament stem cells (PDLSCs) from the damages induced by LPS. However, differently from the present study, RUT at 10 μM had the greater effects on ALP activity and stimulated osteogenic differentiation of PDLSCs [25]. ISO, in this study, significantly increased ALP activity and NM deposition at 50 and 25 μM in MDPC-23 cells. Differently from this study, ISO demonstrated osteogenic differentiation and mineral deposition of MC3T3-E1 and BMSCs at concentrations between 0.1 and 10 μM [21,46].

Considering the results observed, AMP was chosen to be incorporated into PNVCL hydrogel (PNVCL+AMP) and evaluated for cytotoxicity and the effects of hydrogel extracts on mineralization markers in comparison to PNVCL+CH. The PNVCL hydrogel is a thermosensitive polymer capable of acting as a drug delivery system [47]. It has been considered biocompatible when exposed to different types of cells, such as human mesenchymal stem cells (MSCs), encapsulated bovine chondrocytes [30], Caco-2 and pulmonary Calu-3 cell lines [32]. In this study, 48 h and 7-day PNVCL hydrogel extracts with or without AMP or CH were applied for 48 h and evaluated at different time points—48 h, 8 days and 14 days. At low concentrations (dilutions 1/8 to 1/32), PNVCL+AMP and PNVCL+CH extracts were cytocompatible and induced mineralization markers (at dilutions 1/16 and 1/32).

In agreement with our results, other studies have shown cytocompatibility of PNVCL hydrogels [30,33,48]. Injectable thermoresponsive hydroxypropyl guar-graft-poly(N-vinylcaprolactam), or HPG-g-PNVCL, was studied for sustainable release of ciprofloxacin and revealed biocompatibility for being used as a scaffold for osteogenic cell growth [33]. Chondrocytes and mesenchymal stem cells were encapsulated in PNVCL hydrogels and showed cell viability higher than 80%. In this same study, three-dimensional constructs of chondrocytes and PNVCL hydrogels demonstrated time-dependent increases in glycosaminoglycans and collagens in areas of implants in vivo [33]. PNVCL and itaconic acid at 0.2 to 1 mg/mL were synthesized and presented dual pH and temperature-sensitive properties and cytocompatibility in immortalized hepatic cell line Hep G2 [49].

This was the first study evaluating the potential of ampelopsin-loaded PNVCL hydrogels to stimulate biomineralization in odontoblastic cells. Some studies have demonstrated that the therapeutic effects of AMP were enhanced when it was incorporated into different hydrogels [49,50]. AMP (or dihydromyricetin) was incorporated in cationic nanocapsules for safe topical use in photoprotection against UV-induced DNA damage. This system showed 99.9% protection against DNA lesion induction and cytocompatibility when applied to 3T3 mouse fibroblasts, human fibroblasts and HaCaT human epithelial cells (at 10 μg/mL) [49]. Recently, AMP was added to a self-microemulsion delivery system (d-α-tocopheryl polyethylene glycol 1000 succinate-quillaja saponin), and this system enhanced the absorptivity and ability of AMP to prevent hyperlipidemia in mice models [50].

Some limitations of this present study can be pointed out, such as the lack of microscopic analysis to evaluate cell viability; and live/dead staining and some cell adhesion and proliferation fluorescence assays instead of only resazurin-colorimetric assays. Although the release of AMP and CH from PNVLC hydrogels were determined in a previous study [34], the presence of other components of the PNVCL hydrogels and their concentrations in the hydrogel extracts should have been determined in this study.

In view of the results presented, the flavonoid AMP, when incorporated into a PNVCL hydrogel, demonstrated potential for application in endodontic procedures and could overcome the deficiencies of the materials already used—preserving residual cells and increasing their biomineralization ability. For further studies, genetic analysis of specific dentin mineralization markers, such as dentin matrix protein-1 (DMP-1) and dentin sialo phosphoprotein (DSPP), and other osteogenic markers also related to dentinogenesis, such as runt-related transcription factor 2 (RUNX), osteocalcin (OCN) and osteopontin (OPN), will be evaluated. The potential of PNVCL hydrogels as scaffolds for regenerative purposes in endodontics should also be tested. In addition, in vivo studies are necessary to confirm their safety and efficacy for clinical applications.

## 5. Conclusions

AMP was cytocompatible and induced the highest levels of mineralized markers in MDCP-23. Low concentrations of AMP-loaded PNVCL extracts were biocompatible and able to induce ALP activity and mineralized nodule deposition.

## Figures and Tables

**Figure 1 jfb-14-00139-f001:**
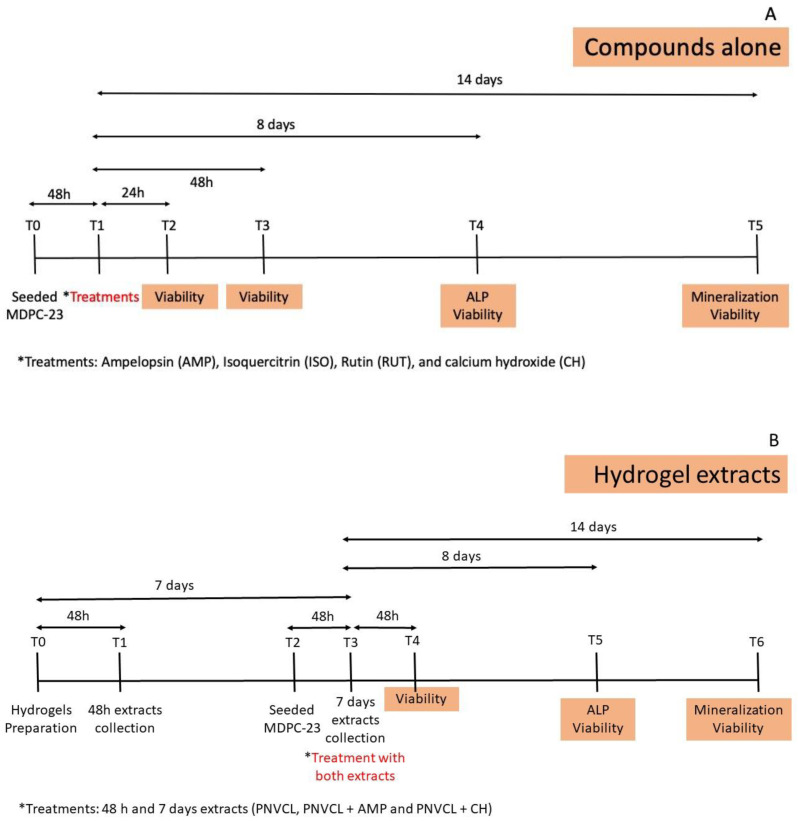
Experimental design of the study. (**A**) MDPC-23 treatments with ampelopsin (AMP), isoquercitrin (ISO), rutin (RUT) and calcium hydroxide (CH). (**B**) MDPC-23 treatments with 48 h and 7 days extracts of the hydrogels: PNVCL, PNVCL+AMP and PNVCL+CH.

**Figure 2 jfb-14-00139-f002:**
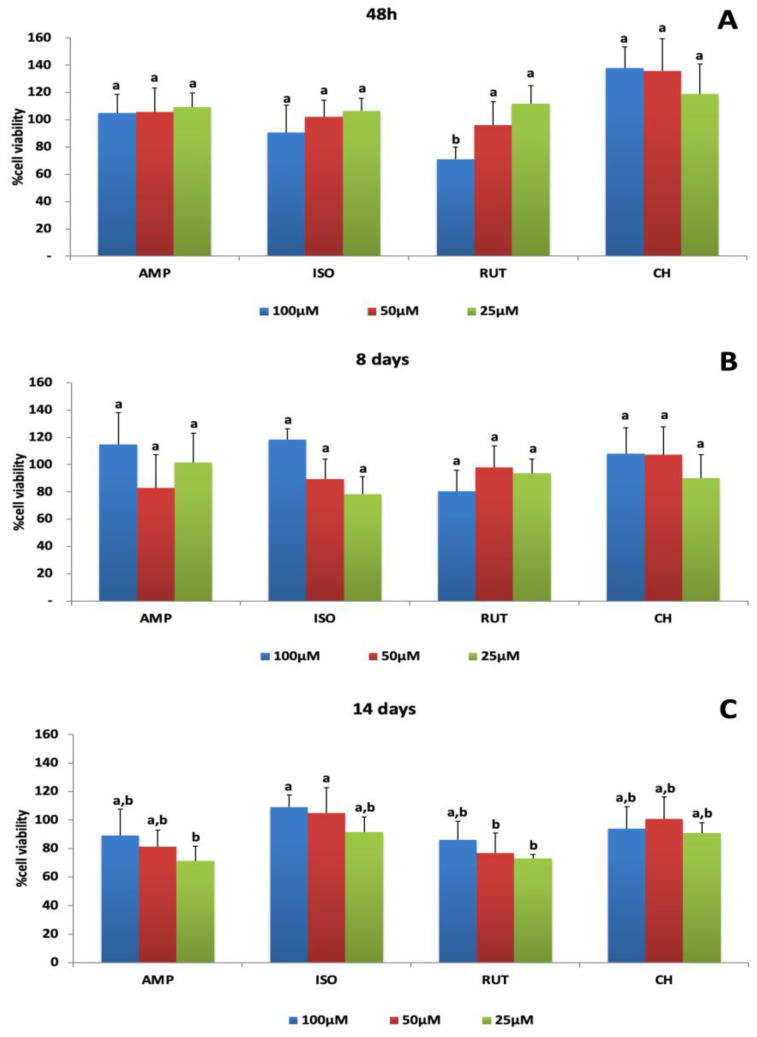
Percentage of MDPC-23 viability after 48 h treatment with ampelopsin (AMP), isoquercetrin (ISO), rutin (RUT) and calcium hydroxide (CH) and evaluation at 48 h (**A**), 8 days (**B**) and 14 days (**C**). The values are expressed as means and standard deviations. ^a,b^ Different letters show statistical differences among the groups and concentrations, according to 2-way ANOVA and Tukey’s post hoc test (*p* < 0.05).

**Figure 3 jfb-14-00139-f003:**
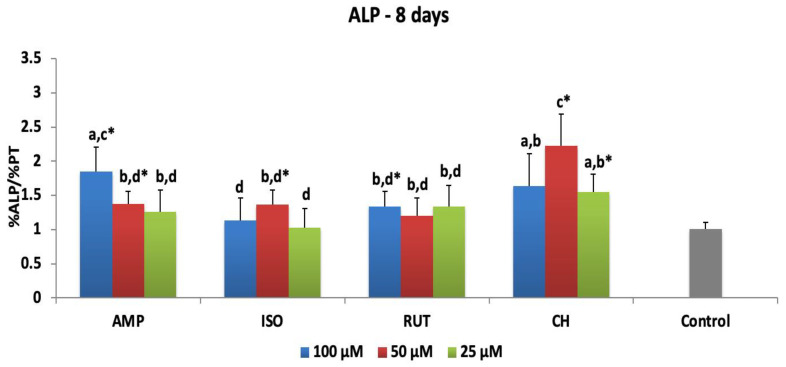
ALP activity of MDPC-23 cells after 48 h of the treatment with ampelopsin (AMP), isoquercetrin (ISO), rutin (RUT) and calcium hydroxide (CH) and 8 days of evaluation. The values are expressed as means and standard deviations. ^a–d^ Different letters show statistical differences among the groups and concentrations, according to 2-way ANOVA and Tukey’s post hoc test (*p* < 0.05). * Statistical difference between each experimental group and the control group (osteogenic DMEM), according to Dunnett test (*p* < 0.05). Grey bar—control osteogenic DMEM.

**Figure 4 jfb-14-00139-f004:**
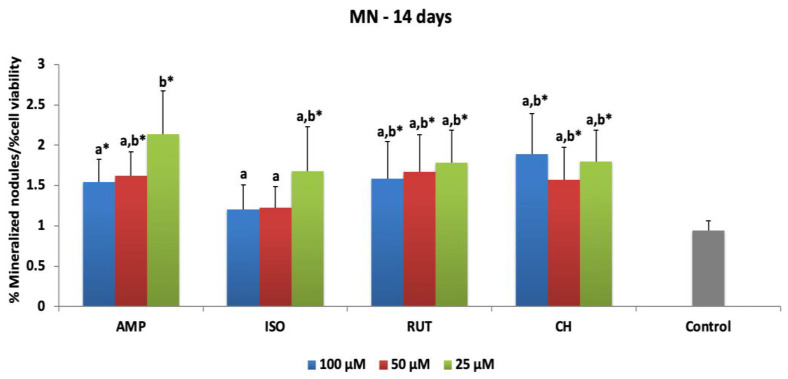
Mineralized nodule deposition of MDPC-23 cells at 14 days of evaluation after 48 h of the treatment with ampelopsin (AMP), isoquercetrin (ISO), rutin (RUT) and calcium hydroxide (CH). The values are expressed as means and standard deviations. ^a,b^ Different letters show statistical differences among the groups and concentrations, according to 2-way ANOVA and Tukey’s post hoc test (*p* < 0.05). * Statistical difference between each experimental group and the control group (osteogenic DMEM), according to Dunnett test (*p* < 0.05). Grey bar—control osteogenic DMEM.

**Figure 5 jfb-14-00139-f005:**
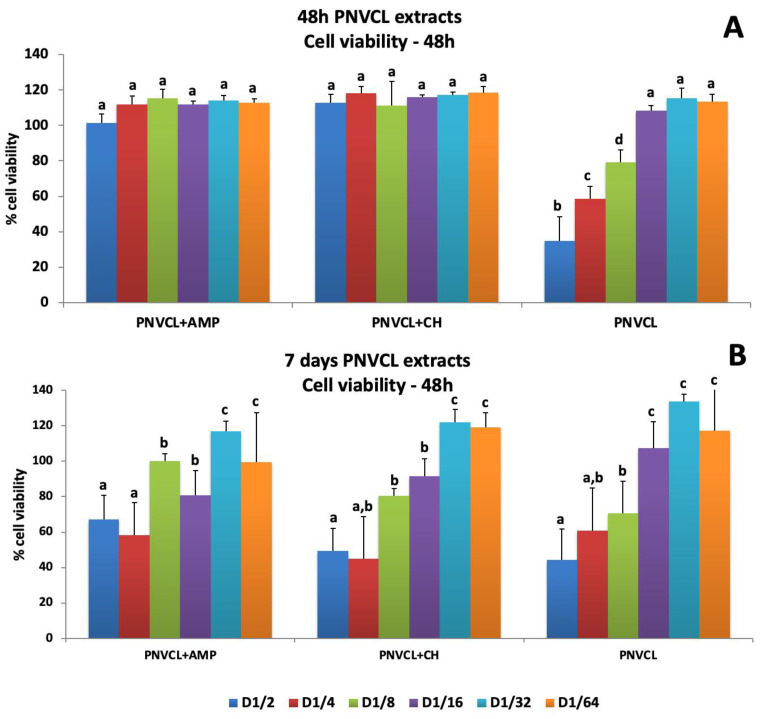
Percentages of MDPC-23 viability after 48 h treatments with 48 h (**A**) and 7-day (**B**) PNVCL hydrogel extracts only, PNVCL with ampelopsin (PNVCL+AMP) and PNVCL with calcium hydroxide (PNVCL+CH). The values are expressed as means and standard deviations. ^a–d^ Different letters show statistical differences among the groups and concentrations, according to 2-way ANOVA and Tukey’s post hoc test (*p* < 0.05). Statistical difference between each experimental group and the control group, according to Dunnett test (*p* < 0.05).

**Figure 6 jfb-14-00139-f006:**
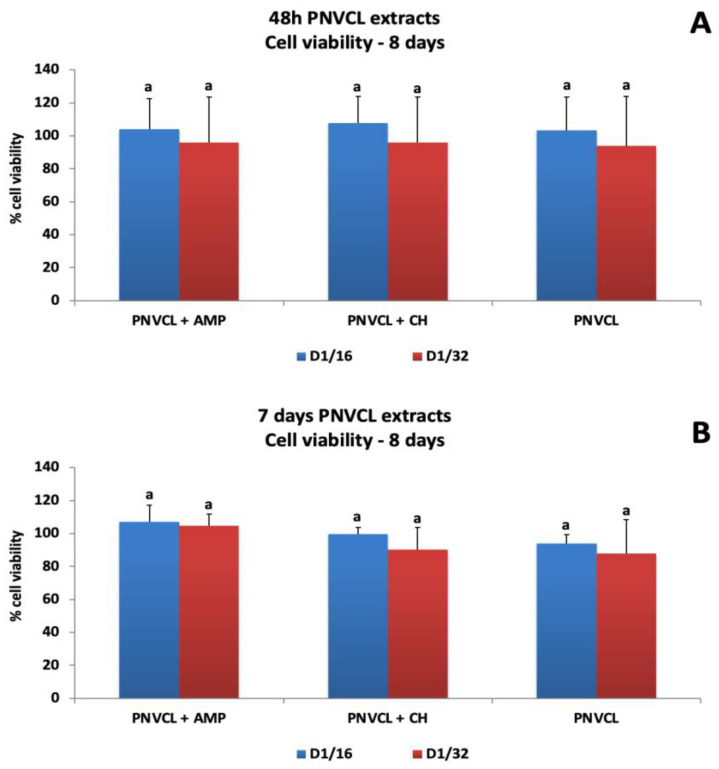
Percentage of MDPC-23 viability after 48 h of treatment with 48 h (**A**) and 7-day (**B**) PNVCL hydrogel extracts with or without ampelopsin (PNVCL+AMP) and calcium hydroxide (PNVCL+CH). Evaluation at 8 days. The values are expressed as means and standard deviations. ^a^ Different letters show statistical differences among the groups and concentrations, according to 2-way ANOVA and Tukey’s post hoc test (*p* < 0.05). Statistical difference between each experimental group and the control group, according to Dunnett test (*p* < 0.05).

**Figure 7 jfb-14-00139-f007:**
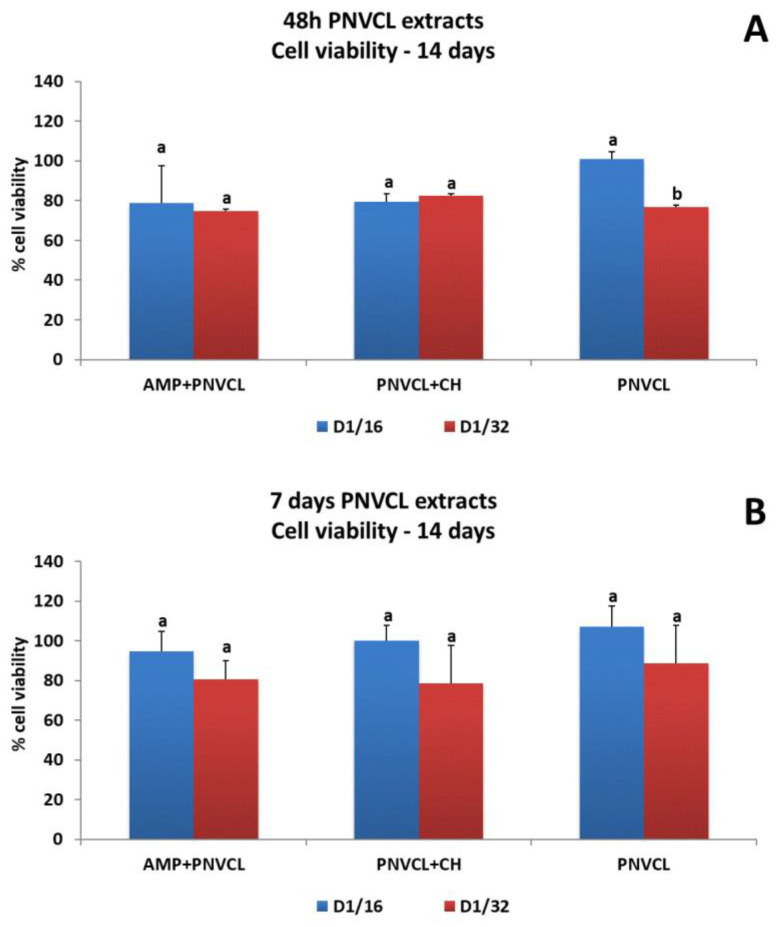
Percentages of MDPC-23 viability after 48 h of treatments with 48 h (**A**) and 7-day (**B**) PNVCL hydrogel extracts with or without ampelopsin (PNVCL+AMP) and calcium hydroxide (PNVCL+CH) and evaluation at 14 days. The values are expressed as means and standard deviations. ^a,b^ Different letters show statistical differences among the groups and concentrations, according to 2-way ANOVA and Tukey’s post hoc test (*p* < 0.05). Statistical difference between each experimental group and the control group, according to Dunnett test (*p* < 0.05).

**Figure 8 jfb-14-00139-f008:**
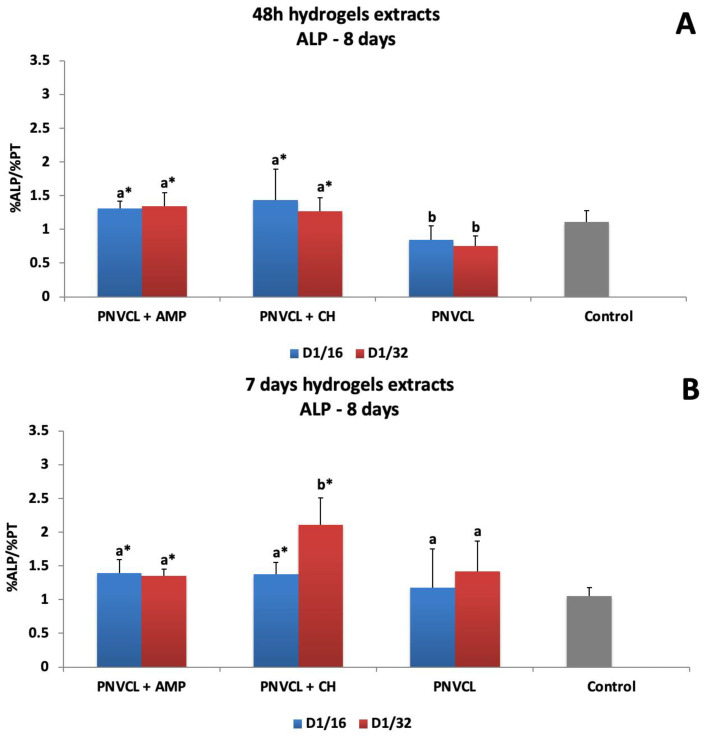
ALP activity of MDPC-23 cells after 48 h treatments with 48 h (**A**) and 7-day (**B**) hydrogel extracts with or without ampelopsin (AMP) and calcium hydroxide (CH). Evaluation at 8 days. ^a,b^ Different letters show statistical differences among the groups and concentrations, according to 2-way ANOVA and Tukey’s post hoc test (*p* < 0.05). * Statistical difference between each experimental group and the control group, according to Dunnett test (*p* < 0.05). Grey bar—control osteogenic DMEM.

**Figure 9 jfb-14-00139-f009:**
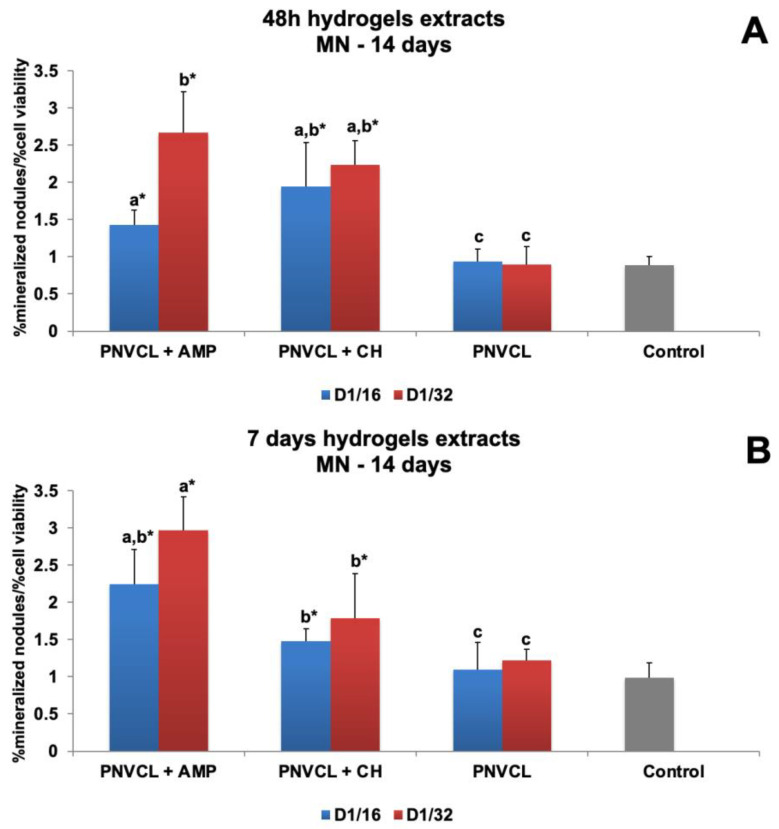
Mineralized nodule deposition by MDPC-23 cells at 14 days of evaluation after 48 h treatments with 48 h (**A**) and 7-day (**B**) hydrogel extracts with or without ampelopsin (AMP) and calcium hydroxide (CH). ^a–c^ Different letters show statistical differences among the groups and concentrations, according to 2-way ANOVA and Tukey’s post hoc test (*p* < 0.05). * Statistical difference between each experimental group and the control group, according to Dunnett test (*p* < 0.05). Grey bar—control osteogenic DMEM.

## Data Availability

Not applicable.

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
