# Peer review of "Cytotoxicity and Biomineralization Potential of Flavonoids Incorporated into PNVCL Hydrogels"

_jfb, 2023, doi:10.3390/jfb14030139_

Round 1

Reviewer 1 Report

Here, the authors evaluated the effects of flavonoids incorporated in poly (N-vinylcapro- 13 lactam) - PNVCL hydrogel on cell viability and mineralization markers of odontoblast-like cells. 14 MDPC-23 cells were exposed to ampelopsin (AMP), isoquercitrin (ISO), rutin (RUT) and control 15 calcium hydroxide (CH) for evaluation of cell viability, total protein (TP) production, Nevertheless, this paper still needs some major revisions then would be accepted.

You have published pervious paper “Microbiological Properties and Cytotoxicity of PNVCL Hydrogels Containing Flavonoids as Intracanal Medication for Endodontic Therapy” 10.3390/jfb13040305.  would you please clearly the aim and purpose of this paper.

I suggest the authors perform Live/Dead staining to evaluate better hydrogel‘ biocompatibility instead of only the resazurin colorimetric assays and the authors should eveloted the adhesion and proliferation cells by labeled by phalloidin-TRITC v

the authors identified the MDPC-23 only using a single marker ALP. It will be better to use more typical osteogenic markers such as Gene expression of ALP, runt-related transcription factor 2 (RUNX), osteocalcin (OCN), and osteopontin (OPN) by using PCR

the MDPC-23 showed to be cultured on the surface of hydrogel instead of inside. It will be more convincing to label the hydrogel and provide more explicit images of 3D reconstruction to prove that the cells are inside the hydrogel.

Author Response

Please see the attachment. Thanks for the careful review.

Reviewer 2 Report

TITLE:  Cytotoxicity and biomineralization potential of flavonoids incorporated into PNVCL hydrogels

The aim of the present investigation was to assess the cytotoxicity and biomineralization potential of odontoblast-like cells stimulated by flavonoids (ampelopsin, isoquercitrin and rutin) and by ampelopsin-loaded PNVCL hydrogel, as a potential injectable delivery system for endodontic purposes.

GENERAL COMMENTS

The article is in-line with the journal topic, but flaws should be improved.  The investigation is interesting, and the present paper is recommended for publication to the present journal after major revision.

Title: The title should indicate the type of study that has been conducted: (f.e.: Cytotoxicity and biomineralization potential of flavonoids incorporated into PNVCL hydrogels: an in vitro study on MDPC-23 odontoblast-like cells.

Introduction

1.      The study model should be supported with an appropriate background section.

2.      Line 36-37: The in vivo osteoinductive properties with dynamic biomineralization processes around these calcium silicate materials extruded in medullary bone (PMID: 28233601)

3.      The ampelopsin (AMP) implications in osteogenesis were not introduced in this section. Please expand this sentence.

4.      The null hypothesis should be added at the end of the introduction section.

Materials and methods

The study samples should be declared in this section.

Fig. 3 Statistical difference from each experimental group and the control group (osteogenic DMEM), according to Dunnett test (p<0.05).

The authors declared a different post-hoc test. This part should be moved to statistical methods section.

Results

The figures S1, S2 and S3 are too low in quality, please improve this items.

Discussion

“Alkaline phosphatase (ALP) activity is the initial phase of the dentin matrix biomineralization and the formation of mineralization nodules is the final product of cell differentiation”.

In my opinion, the alkaline phosphatase (ALP) activity should be considered as ONE of the bioactivity marker, but an analysis of genes expression (RT-PCR) is necessary to determine a specific pathway. The author could discuss this aspect as a potential limit of the study.

The null-hypothesis should be discussed in this part of the manuscript.

Author Response

(The authors gave the same response as above.)

Reviewer 3 Report

1) Which are the components of the PNVCL hydrogel that can be found in the extract and in which concentration? And why did the authors choose to work with hydrogel extract and not with PNVCL hydrogel in direct contact with cells, if the goal is to use PNVCL as a potential injectable delivery system of flavonoids for endodontic purposes?

2) I suppose that at on line 318 the authors also refer to hydrogel extract ("pure PNVCL hydrogel....").

3) For a better reading and understanding, the explicative text and numbers from all figures could be slightly increased.

Author Response

(The authors gave the same response as above.)
